# High Hydration Factor in Older Hispanic-American Adults: Possible Implications for Accurate Body Composition Estimates

**DOI:** 10.3390/nu11122897

**Published:** 2019-11-29

**Authors:** Rogelio González-Arellanes, Rene Urquidez-Romero, Alejandra Rodríguez-Tadeo, Julián Esparza-Romero, Rosa-Olivia Méndez-Estrada, Erik Ramírez-López, Alma-Elizabeth Robles-Sardin, Bertha-Isabel Pacheco-Moreno, Heliodoro Alemán-Mateo

**Affiliations:** 1Centro de Investigación en Alimentación y Desarrollo, A.C. Coordinación de Nutrición. Carretera Gustavo Enrique Astiazarán Rosas #46, Col. La Victoria C.P. 83304, Hermosillo, Sonora, Mexico; rogelio.gonzalez@estudiantes.ciad.mx (R.G.-A.); julian@ciad.mx (J.E.-R.); romendez@ciad.mx (R.-O.M.-E.); berthai@ciad.mx (B.-I.P.-M.); 2Universidad Autónoma de Ciudad Juárez. Instituto de Ciencias Biomédicas. Departamento de Ciencias de la Salud. Ave. Plutarco Elías Calles #1210, Col. Fovissste Chamizal C.P. 32310, Ciudad Juárez, Chihuahua, Mexico; rurquide@uacj.mx (R.U.-R.); alrodrig@uacj.mx (A.R.-T.); 3Universidad Autónoma de Nuevo León. Facultad de Salud Pública y Nutrición. Ave. Dr. Eduardo Aguirre Pequeño #905, Col. Mitras Centro C.P. 64460, Monterrey, Nuevo León, Mexico; erik.ramirezl@uanl.mx

**Keywords:** aging, body composition, obesity, hydration factor and Hispanic Americans

## Abstract

Age- and obesity-related body composition changes could influence the hydration factor (HF) and, as a result, body composition estimates derived from hydrometry. The aim of the present study was to compare the HF in older Hispanic-American adults to some published values. This cross-sectional study included a sample of 412 subjects, men and women, aged ≥60 years from northern Mexico. HF values were calculated based on the ratio of total body water-using the deuterium dilution technique-to fat-free mass, derived from the four-compartment model. The mean HF value for the total sample (0.748 ± 0.034) was statistically (*p* ≤ 0.01) higher than the traditionally assumed value of 0.732 derived from chemical analysis, the “grand mean’’ value of 0.725 derived from in vivo methods, and the 0.734 value calculated for older French adults via the three-compartment model. The HF of the older women did not differ across the fat mass index categories, but in men the obese group was lower than the normal and excess fat groups. The hydration factor calculated for the total sample of older Hispanic-American people is higher than the HF values reported in the literature. Therefore, the indiscriminate use of these assumed values could produce inaccurate body composition estimates in older Hispanic-American people.

## 1. Introduction

The demographic and epidemiological transition represents a huge challenge for governments and health and aging institutions worldwide, especially the double burden of malnutrition in several age groups, including older people [1,2]. Appropriate nutritional assessments of older people are necessary to improve medical and nutritional management and to design appropriate interventions. In this regard, accurate tools for assessing body composition in older people are key. Following Penchard and Azcue [3], fat-free mass (FFM) and fat mass (FM) are the most appropriate parameters for defining malnutrition. The most common methods used to assess body composition include such two-compartment (2C) models as hydrometry, among others. Specifically, this method calculates FFM in kilograms (kg) based on the ratio of total body water (TBW) to the hydration factor value [4]. Subsequently, FM can be estimated by the difference in body weight, measured in kg, and FFM, also measured in kg. The hydration factor (HF) is the ratio of TBW in kg to FFM solids in kg, and its magnitude, according to chemical analyses, is around of 0.730 [4]. That HF value was obtained from ten human cadavers (seven males, three females) aged 25–67 years. The HF ranges in non-oedematous and oedematous corpses were 0.686–0.776 and 0.808–0.824, respectively [5,6,7,8,9,10]. New HF values obtained with two and three-compartment models have been reported. Ritz et al. [11] estimated a “grand mean” hydration factor of 0.725, compiled from data in several studies based on in vivo methods. Those researchers also found that the HF is similar between young (*n* = 35, 0.732 ± 0.024) and older subjects (*n* = 68, 0.734 ± 0.024).

In the field of body composition, a HF of 0.732 is generally accepted as a constant throughout the life-cycle, regardless of factors such as training level, obesity, age, gender, and ethnicity [12,13,14]. Regarding age, older subjects (60–69 years) had lower mean TBW values (35.4 liters) at the molecular level than adults (20–29 years; 39.1 liters) [15]; however, it seems that the HF estimated by in vivo methods is not affected by aging [11,16]. Obese subjects, meanwhile, could have a different HF than non-obese subjects, regardless of age and gender. Fuller et al. [17] found a low HF of 0.712 ± 0.016 with a range of 0.682–0.751 in obese adult women, whereas Das [18] found a high HF (0.756) in extremely obese men and women subjects, and Haroun et al. [19] reported a higher HF in obese children (0.763) than the mean value for non-obese children (0.737). Ritz et al. [20] reported a lower HF in obese adult women (0.733 ± 0.023) than in lean adult women (0.747 ± 0.022), although the HF was significantly higher (0.768 ± 0.012) in obese than lean adult men (0.740 ± 0.019). These results show that the behavior of the HF in conditions of obesity is not yet clear, and that our knowledge of aged and obese people in this regard is insufficient. If HF differs from the assumed value of 0.732 due to factors such as age, obesity, and ethnicity, then estimates of body composition may be affected and may lead to erroneous associations and misclassifications of people’s nutritional status.

All the studies cited above [17,18,19,20] used the body mass index (BMI) to diagnose obesity, but this parameter is an unspecific marker of adiposity and, therefore, inaccurate for diagnosing obesity [21]. The fat mass index (FMI), in contrast, is a gender-specific measure of fat that is not confounded by lean tissue. Hence, it is one of the most accurate approaches available for diagnosing obesity [22]. In summary, the HF in obese older adults is still unknown. Therefore, the aims of the present study were to determine the HF in older Hispanic-American adults, and then compare it to some published values. Differences in the HF between obese and non-obese older Hispanic-American adults were also explored.

## 2. Materials and Methods

### 2.1. Subjects

This study included a sample of community-dwelling older (≥60 years) Hispanic-American adults from northern Mexico with stable body weight-by self-report-in the 3 months prior to the study. All subjects were apparently healthy and physically-independent according to Lawton and Brody’s scale [23]. None were involved in sports activities. According to clinical assessment, all volunteers were free of severe oedema, medication, diseases, and metabolic disorders, such as cancer, diabetes, heart disease, and kidney or liver failure, that could have affected their hydration status and body composition. Subjects with well-controlled hypothyroidism and hypertension were included. All participants underwent an oral glucose tolerance test to confirm that they were free of type-2 diabetes and their specific gravity urine and haematocrit values indicated a good hydration status.

### 2.2. Design

A cross-sectional design was implemented. All volunteers were recruited from 2016–2019 in Hermosillo, Sonora; Ciudad Juárez, Chihuahua; and Monterrey, Nuevo León. The study protocol included two visits to the Body Composition Laboratories by all test subjects. During the first visit, potential volunteers underwent clinical and lab assessment to apply the inclusion and exclusion criteria. On the second, all the volunteers who met the inclusion criteria underwent anthropometric and body composition measurements. The study protocol was approved by the Ethics Committee of the Centro de Investigación en Alimentación y Desarrollo, AC (CE/008/2014), the Universidad Autónoma de Ciudad Juárez (CBE.ICB/023.10–14), and the Universidad Autónoma de Nuevo León (15-FaSPyN-SA-19). Volunteers received a full explanation of the protocol and signed the appropriate informed consent. 

### 2.3. Anthropometry

All anthropometric variables were measured using a standard methodology [24]. Weight in kg was measured to the nearest 0.1 kg using an electronic scale (SECA 878, Germany, with the scale attached to the Bod-Pod System). Height in meters (m) was measured to the nearest 0.1 centimeter (cm) using a stadiometer (SECA 264, Germany). BMI was obtained from the weight in kg (SECA scale) divided by height-squared in m.

### 2.4. Body Composition and HF Measurements

In order to calculate the HF, fat-free mass was derived from the four-compartment (4C) model, which requires the following three independent measurements: 

(1) Total body water. This is the major component of FFM. It was measured using deuterium oxide (D_2_O, 99.8 atom percent, Lot. No. 14G-316, Cambridge Isotope Laboratories, Inc., USA) by two protocols. In one laboratory, D_2_O was first measured by isotope ratio mass spectrometry (IRMS), though this was later replaced by a Fourier transform infrared spectrophotometer (FTIR; 8400S, Cat No. 206-72400-92, Shimadzu Corporation, USA). The FTIR technique was chosen because this equipment became available in the laboratory. It is important to note that the results of D_2_O quantification in saliva samples by IRMS and FTIR did not differ [25]. TBW measurements were made in accordance with the International Atomic Energy Agency [14]. 

(2) Bone mineral content. The densest body composition component, or bone mineral content (BMC), was measured by dual-energy X-ray absorptiometry (DXA) using a General Electric Lunar DPX-MD+ at the Centro de Investigación en Alimentación y Desarrollo A.C., by the Lunar iDXA at the Universidad Autónoma de Nuevo León, and by Lunar prodigy advance at the Universidad Autónoma de Ciudad Juárez, following the previously published procedure [26] at each participating institution. Total body mineral mass (TBMM) was calculated using the following equation; TBMM=BMC×1.279, a factor that is an assumed value which represents the sum of osseous and cell mineral content [27]. For the present study, all DXA scans collected in each laboratory were edited by a trained staff member using LU43616ES©2015 GE Healthcare Lunar encore software. In addition, we ensured that BMC was measured in all obese and extremely obese subjects. The DXA equipment was calibrated daily in accordance with the manufacturer’s guidelines before taking measurements.

(3) Body density. This variable was determined by the air displacement plethysmography technique using the Bod-Pod system (BodPod® Body Composition System, Life Measurement Instruments, Concord, CA, USA), following the protocol reported previously [26]. For the present study, total body volume (TBV) was corrected by thoracic gas volume (TGV), which was measured in most, but not all, subjects, since a few were unable to fulfil the requirements for measuring this factor. In those cases, the TGV predicted by the system was used to correct TBV, considering a determination coefficient of 0.96 between FM by the 4C model, and using the BD with TBV corrected by both measured and estimated TGV. The Bod-Pod system was calibrated daily in accordance with the manufacturer’s guidelines.

First, these individual body composition components were used to obtain the corresponding fraction of each component to body weight, measured by the scale attached to the Bod-Pod System. The aqueous weight fraction (A) was obtained from the ratio of TBW in kg to body weight in kg, while the mineral weight fraction (M) was calculated from the following TBMM relations in kg/body weight in kg. After that, the body fractions, together with BD, were incorporated into Baumgartner’s equation [27] to estimate the fat mass percentage, $\%FM=205\times \left(\frac{1.34}{BD}-0.35\left(A\right)+0.56\left(M\right)-1\right)$ Subsequently, FM in kg was calculated with the following equation: $FM=\frac{\%FM\times weight}{100}$. Finally, FFM in kg was obtained from the differences between body weight in kg and FM in kg.

### 2.5. Hydration Factor

The HF was calculated as the ratio of TBW in kg, derived from the deuterium dilution technique, to FFM in kg, derived from the 4C model, such that $HF=\frac{TBW}{FFM}$.

### 2.6. Obesity Classification

In addition, the FMI was obtained using the 4C model. FM in kg derived from this model was divided by height-squared in m. Three classification ranges were obtained, as follows: Normal (3.0 to 6.0 kg/m^2^ and 5.0 to 9.0 kg/m^2^ for men and women, respectively), excess fat (>6.0 to 9.0 kg/m^2^ and >9.0 to 13.0 kg/m^2^ for men and women, respectively), and obesity (>9.0 kg/m^2^ and >13.0 kg/m^2^ for men and women, respectively) [22].

### 2.7. Statistical Analyses

The gender differences for several variables in the FMI categories were tested by a two-sample independent *t*-test. The differences for several variables within each gender across the FMI classification ranges were then tested by a one-way analysis of variance (ANOVA) with a post-hoc Tukey test (*p* ≤ 0.05). The mean HF value obtained for the total sample, obese, and normal older subjects was compared to (i) the traditionally assumed value of 0.732, derived from chemical analyses [12]; (ii) the “grand mean” value of 0.725, derived from in vivo methods [11]; and (iii) the 0.734 value, derived from 68 healthy, non-obese, older French people [11] by separate analyses using the one-sample *t*-test. Significance was considered at a *p*-value ≤ 0.05. All analyses were run in the STATA/SE 12.0 statistical program (StataCorp LP, TX, USA). 

## 3. Results

A total sample of 412 (265 women, 147 men) older Mexican subjects, aged 60–90 years, with BMIs in the range of 18.7–43.6 kg/m^2^-which corresponds to an FMI range of 3.6–24.7 kg/m^2^-were included. Based on the FMI ranges, 10.7%, 46.8%, and 42.5% subjects were classified as normal, with excess fat, and obese, respectively. 

As expected, there was a difference between gender, regardless of FMI category, on some of the anthropometric variables analyzed by the two-sample independent *t*-test, as the men had significantly higher body weight and height than the women. FFM was also significantly higher in men than women (*p* ≤ 0.05). In contrast, the mean FM and FMI values were higher in women than men (*p* ≤ 0.05). According to the results of the ANOVA and post-hoc Tukey test, the differences in anthropometric and body composition variables within gender across the FMI categories shows that the mean values for body weight, BMI, FM, and FMI in the obesity category were higher than in the normal and excess fat category in both men and women (Table 1). 

Regarding the HF, Figure 1 shows the correlation between FM and the components of the HF by Pearson’s correlation test. It seems that in both genders there is a positive and moderate correlation (*p* < 0.001) between FM and TBW and FFM. However, the results in Table 1 show that a between-gender difference also appeared, as the mean HF value in men was lower than in women, though only in the obesity category. With respect to the behavior of the HF by FMI category, Table 1 shows that the mean HF value was lower in the men’s obesity category than in the normal and excess fat categories. In the women’s group, in contrast, the mean HF values were close across all FMI categories.

Table 2 shows the results of the comparison between the HF calculated in this sample of older Hispanic-American adults to the values published in the literature (calculated mostly from Caucasian populations) by the one-sample *t*-test. The mean HF value for the total sample, i.e., for the normal, excess fat, and obesity categories together and separate, was statistically (*p* ≤ 0.01) higher than the 0.732 value derived from chemical analyses, the “grand mean’’ value of 0.725 derived from in vivo methods, and the 0.734 value derived from older French adults using a multi-compartment model.

## 4. Discussion

This is the first study to assess the HF in a wide sample of older Hispanic-American adults with one of the most highly-recommended models for accurately determining fat mass and, hence, FFM; that is, the 4C model. This model considers the main molecular changes in body composition across the normal aging process, as well as obesity, bone mineral content, and TBW. In addition, TBW was assessed by the gold standard methodology using the deuterium dilution technique, following the recommended protocols. Therefore, the results for the HF estimated using these high-standard methodologies should raise awareness regarding the accuracy and precision of the hydrometric method for assessing body composition in older obese non-Caucasian subjects. 

The accuracy of the hydrometric method primarily depends on obtaining an adequate value for the HF in order to estimate body composition, since these estimates could prove to be inaccurate if the assumed HF value differs from the “real” value [12]. Another often neglected contributing factor to hydrometric inaccuracy is the TBW determination. Unfortunately, variations in estimates of the HF value are common, likely due to differences among the methods used to measure both FFM and TBW. Two of the most important limitations of the chemical analysis of cadavers in terms of assessing hydration for the FFM is that the bodies (i) were analysed post-mortem, and (ii) that several subjects had suffered severe illnesses before death. Both factors could affect hydration status. Another significant limitation is the insensible water loss between the time of death and the performance of the chemical analysis [4].

Despite these limitations, the HF derived from the analysis of cadavers is one of the values that is most often used in the literature to estimate FFM. Our results clearly show a high HF for the total sample of older men and women subjects, compared to 0.732 (Table 2), which was independent of the FMI range classifications. We cannot, therefore, ignore the possible effect of the methods chosen. The HF calculated in the present study was consistently higher than the aforementioned 0.732 derived from chemical analysis, 0.725 derived from two-compartment model, and 0.734 values derived from three-compartment model. In addition, we cannot ignore a possible effect of ethnicity on the TBW and FFM variables. That older French (*n* = 68) subjects had a higher mean FFM in kg (46.1 ± 1.1) than our older Hispanic-American subjects (43.5 ± 8.9), though their mean TBW values in kg (33.9 ± 0.8 and 32.6 ± 6.8, respectively) were similar. Therefore, the ratio of TBW to FFM in older French adults is lower than in our older Hispanic-American adults. It should be noted that the effect of ethnicity on body composition in older people is well-recognized [28].

To speculate on the possible effect of methods on HF differences, we compared the HF calculated in the present study to values reported for older people using the 4C model to estimate FFM and TBW by deuterium dilution. Here, our HF results (0.748 ± 0.033) were similar to the mean HF values reported by Baumgartner et al. [27] (0.744) and Alemán-Mateo et al. [26] (0.752) for Caucasian and non-Caucasian older adults, respectively using Baumgartner’s equation. In addition, using Baumgartner’s equation, Goran et al. [29] found a higher value (0.747) of HF in older men, but not in women. In contrast, [30] using the 4C model, particularly Selinger’s equation, Yee et al. found a higher value (0.761) in older women. It is important to clarify that these authors did not compare their results statistically against HF values reported in the literature. Considering these findings, it seems that there is only an effect of method and aging on the HF for the total sample assessed; however, additional studies of this kind are required in order to reach more general conclusions regarding the effect of methods and aging on the HF.

With respect to the influence of adiposity on the HF, existing evidence is limited and unclear, perhaps because this phenomenon could not be explained by certain physiological or biochemical mechanisms. Specifically, the mean HF value of the men in our obesity category (0.737 ± 0.033) was lower than for those in the normal category (0.759 ± 0.025), but similar to the 0.732 value (Table 1) derived from chemical analysis. According to our data, the mean TBW and FFM values of the men in the obesity category (39.3 ± 5.9 kg and 53.3 ± 7.7 kg, respectively) were higher than those of the normal category (35.7 ± 3.7 kg and 47.1 ± 5.3 kg, respectively). These differences in the mean TBW and FFM values led to a decrease in the HF (0.737 ± 0.033) values of the men in the obesity category compared to the HF values for those in the normal category (0.759 ± 0.025). In addition, the mean TBW and FFM values showed a different behaviour by gender across all FMI categories. In the women’s group, mean TBW and FFM values were higher in the obesity than the normal category, though this did not affect HF values. In the men’s group, we observed a slight decrease in TBW with a slight increase in FFM (*p* > 0.05) in the obesity category, compared to the excess fat category. These small changes decreased the HF values of the men in the obesity category.

Considering the present results, the hydrometric methods that assume a HF of 0.732 may be inadequate for obtaining accurate and precise body composition estimates in older Hispanic-American adults with a wide range of FMI. If we were to calculate the body composition in our sample using the hydrometric method and an HF of 0.732, we could overestimate the FFM and, therefore, underestimate the FM. For example, a subject with 70 kg of body weight and 31 kg of TBW would have an FFM (TBW/0.732) of 42.3 kg and an FM of 27.7 kg. However, applying our mean HF value of 0.748 decreased the FFM while increasing the FM by around 1 kg in both cases (FFM = 41.4 kg and FM = 28.6 kg). The inaccuracy of the hydrometric method due to the use of an erroneous HF value could appear in studies of other ethnic groups. Therefore, we highly recommend conducting studies designed to validate the hydrometric method in older adults from various ethnic groups.

## 5. Conclusions

The hydration factor in the total sample was higher than the classic value and the values calculated using in vivo methods in young Caucasian adults and older Caucasian people. However, other researchers using the 4C model in older adults have reported similar HF values, highlighting that older populations possibly have higher values of HF than young adult populations. The men in the obesity category in our study had a lower HF than those in the normal and excess fat categories, with a value similar to 0.732. Therefore, these assumed values may be inadequate for use with older Hispanic-American people in terms of accurately and precisely assessing body composition estimates.

## Figures and Tables

**Figure 1 nutrients-11-02897-f001:**
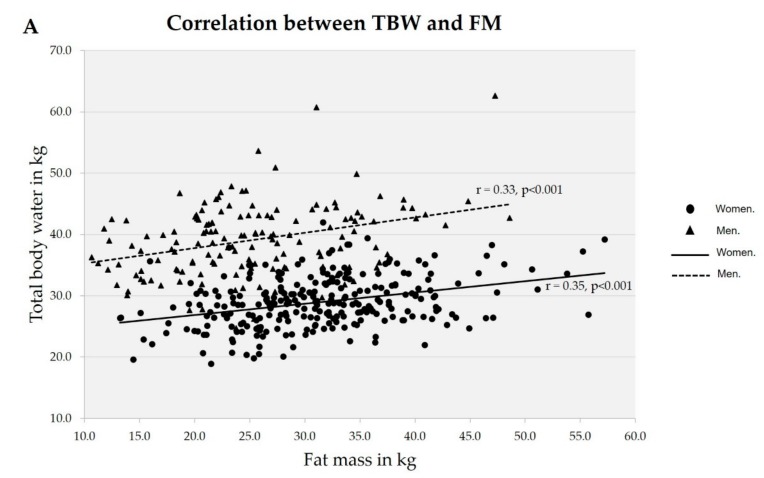
Correlation between total body water (**A**), fat-free mass (**B**), and hydration factor (**C**) to fat mass by gender.

**Table 1 nutrients-11-02897-t001:** Age, anthropometric, and body composition characteristics, and the FMI range classification assumed by gender.

Variable	Normal	Excess fat	Obesity	Total
Men				
Age, years	71.3 ± 5.5 ^a^	67.9 ± 6.1 ^a^	67.7 ± 5.8 ^a^	68.3 ± 5.9
Weight, kg	60.1 ± 5.3 ^a,^*	75.2 ± 9.5 ^b,^*	85.7 ± 11.9 ^c,^*	77.7 ± 13.1 *
Height, m	1.7 ± 0.04 ^a,^*	1.7 ± 0.07 ^a,^*	1.7 ± 0.06 ^a,^*	1.7 ± 0.06 *
BMI, kg/m^2^	21.8 ± 1.9 ^a^	26.3 ± 2.1 ^b^	29.7 ± 3.2 ^c^	27.1 ± 3.7
FFM, kg	47.1 ± 5.3 ^a,^*	53.1 ± 7.8 ^b,^*	53.3 ± 7.7 ^b,^*	52.4 ± 7.7 *
TBW, kg	35.7 ± 3.73 ^a,^*	39.9 ± 6.0 ^b,^*	39.3 ± 5.9 ^b,^*	39.1 ± 5.8 *
FM, kg	13.8 ± 1.7 ^a^	22.1 ± 2.9 ^b^	32.4 ± 5.8 ^c^	25.3 ± 7.9
FMI, kg/m^2^	4.9 ± 0.7 ^a^	7.7 ± 0.8 ^b^	11.2 ± 1.6 ^c^	8.8 ± 2.5
HF	0.759 ± 0.025 ^a^	0.752 ± 0.027 ^a^	0.737 ± 0.033 ^b^	0.746 ± 0.030
Women				
Age, years	69.1 ± 8.3 ^a^	67.9 ± 6.7 ^a^	69.2 ± 6.4 ^a^	68.5 ± 6.8
Weight, kg	55.2 ± 6.7 ^a^	65.9 ± 7.5 ^b^	77.5 ± 8.9 ^c^	69.9 ± 10.8
Height, m	1.6 ± 0.07 ^a^	1.5 ± 0.07 ^a^	1.5 ± 0.06 ^a^	1.6 ± 0.06
BMI, kg/m^2^	22.9 ± 1.8 ^a^	27.2 ± 2.1 ^b,^*	32.4 ± 3.1 ^c,^*	29.0 ± 4.1 *
FFM, kg	36.6 ± 5.1 ^a^	38.3 ± 5.1 ^a,b^	39.5 ± 4.6 ^b^	38.6 ± 4.9
TBW, kg	27.2 ± 4.3 ^a^	28.6 ± 4.2 ^a,b^	29.7 ± 3.9 ^b^	28.9 ± 4.1
FM, kg	18.7 ± 2.9 ^a,^*	27.7 ± 3.6 ^b,^*	37.9 ± 5.9 ^c,^*	31.3 ± 7.9 *
FMI, kg/m^2^	7.8 ± 1.0 ^a,^*	11.4 ± 1.1 ^b,^*	15.9 ± 2.3 ^c,^*	12.9 ± 3.2 *
HF	0.743 ± 0.048 ^a^	0.746 ± 0.033 ^a^	0.753 ± 0.034 ^a,^*	0.748 ± 0.035

BMI = body mass index, FFM = fat-free mass, TBW = total body water, FM = fat mass, FMI = fat mass index, HF = hydration factor. * *p* < 0.05 the between-gender comparison in each FMI category was tested by an two-sample independent *t*-test. ^a,b,c^
*p* < 0.05 the differences within each gender across the FMI classification ranges were tested by a one-way analysis of variance with a post-hoc Tukey test.

**Table 2 nutrients-11-02897-t002:** Comparison of the mean hydration factor values in older non-Caucasian adults to several hydration factors cited in the literature.

FMI Category	Mean	0.732 Value from Chemical Analysis	0.725 Value from “Grand Mean”	0.734 Value from Older French Adults
Normal (*n* = 44)	0.750 ± 0.039	*p* = 0.003	*p* ≤ 0.001	*p* = 0.008
Excess fat (*n* = 193)	0.748 ± 0.031	*p* ≤ 0.001	*p* ≤ 0.001	*p* ≤ 0.001
Obesity (*n* = 175)	0.747 ± 0.035	*p* ≤ 0.001	*p* ≤ 0.001	*p* ≤ 0.001
Total sample (*n* = 412)	0.748 ± 0.033	*p* ≤ 0.001	*p* ≤ 0.001	*p* ≤ 0.001

FMI = fat mass index; comparison between mean hydration factor values and several hydration values cited in the literature by a one-sample *t*-test.

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
