# Peer review of "High Hydration Factor in Older Hispanic-American Adults: Possible Implications for Accurate Body Composition Estimates"

_nutrients, 2019, doi:10.3390/nu11122897_

Round 1

Reviewer 1 Report

Dear Authors,

The different sections of the manuscript are well written, well done.

The abstract section provides a good overview of the paper. 

Line 20, I suggest inserting 'to'in between 'was compared''. It should now read: ''was to compare''

The introductory section gives the reader a general background of the study but the rationale needs to be explained better. A sentence more to explain why it is important to get this right in older adults will be useful.

The method is well written but it will benefit from a clearer description of the inclusion and exclusion criteria.

The design is appropriate, ethics approval obtained and statistical analysis well explained.

Results and discussion, well integrated and the tables and figures are clearer written and well described.

Author Response

Please, see the attach. Thanks

Reviewer 2 Report

The paper reports body composition measures in Mexican older people with a good methodology and the paper is well written. The introduction provides a very clear state of the art about the HF. my main question is "what can explain different HF values with increased adiposity in men and women ?". This point was unclear for me and at the end of the paper I don't understand better.

One of the main argument was the ethnicity. I appreciate the word racial was not used but behind ethnicity we can suppose to find clusters of particular phenotypes. For instance visceral and subcutaneous adiposity may differ according to gender, age and population clusters. Have the authors recorded data about viscreal obesity ? French people are not all caucasian and caucasian people are very heterogeneous, particularly in term of obesity distribution and environmental factors accross Europe or USA.

Thus we need to read hypothesis other than population clusters to explain the possible factors associated to change in HF for the interest of other people than Mexicans.

Minor remarks

line 170 I don't understand this sentence (and the result)"Table 1 shows that the mean HF value was lower in the obesity category than the mean HF value for the normal and excess fat categories. "

Table 1: it seems that several footnote are lacking to explain the a b, c and also I don't understand the meaning of the stars in the headings of the lines.

Author Response

Please see the attach file
